# mRNA Detection in Anal Cytology: A Feasible Approach for Anal Cancer Screening in Men Who Have Sex with Men Living With HIV

**DOI:** 10.3390/diagnostics9040173

**Published:** 2019-11-02

**Authors:** Marta del Pino, Cristina Martí, Jina Gaber, Cecilia Svanholm-Barrie, Leonardo Rodríguez-Carunchio, Adriano Rodriguez-Trujillo, Núria Carreras, Irene Fuertes, Esther Barnadas, Lorena Marimón, Jose Luis Blanco, David H. Persing, Aureli Torné, Jaume Ordi

**Affiliations:** 1Institute Clinic of Gynaecology, Obstetrics, and Neonatology, Hospital Clínic—Institut d’Investigacions Biomèdiques August Pi i Sunyer (IDIBAPS), University of Barcelona, 08036 Barcelona, Spain; marti@clinic.cat (C.M.); adrodrig88@gmail.com (A.R.-T.); ncarreras@clinic.cat (N.C.); atorne@clinic.cat (A.T.); 2Cepheid AB, 71 54 Solna, Sweden; jina.gaber@cepheid.se (J.G.); cecilia.svanholmbarrie@cepheid.se (C.S.-B.); 3Department of Pathology, Hospital Clínic, University of Barcelona, 08036 Barcelona, Spain; lerodrig@clinic.cat (L.R.-C.); esther.barnadas@idibaps.org (E.B.); lmarimon@clinic.cat (L.M.); jordi@clinic.cat (J.O.); 4Department of Dermatology, Hospital Clinic of Barcelona, University of Barcelona, 08036 Barcelona, Spain; ifuertes@clinic.cat (I.F.); JLBLANCO@clinic.cat (J.L.B.); 5Cepheid AB, Sunnyvale, CA 94089, USA; david.persing@cepheid.com; 6Institut de Salut Global de Barcelona (ISGlobal), 08036 Barcelona, Spain

**Keywords:** anal cancer, anal cytology, HPV, mRNA, biomarkers

## Abstract

There is growing interest in anal cancer screening strategies. However, cytological/molecular evaluation of anal samples is challenging. We aimed to determine the feasibility of detecting, in anal liquid-based cytologies, the expression of biomarkers involved in the cell cycle disturbance elicited by human papillomavirus (HPV). The accuracy of this approach in the identification of high-grade squamous intraepithelial lesions/anal intraepithelial neoplasia grade2–3 (HSIL/AIN2–3) was also evaluated. 215 anal cytologies from men having sex with men living with human immunodeficiency virus were evaluated. Patients showing concordant cytological and anoscopy-directed biopsy diagnosis were selected: 70 with negative cytology and HPV test, 70 with low-grade SIL (LSIL/AIN1) cytology and biopsy, and 75 with cytology and biopsy of HSIL/AIN2–3. CDKN2A/p16, MKI67 and TOP2A mRNA expression was analyzed. HPV detection was performed with Xpert HPV Assay (Cepheid, Sunnyvale, CA, USA). HSIL/AIN2–3 showed higher expression for the biomarkers than LSIL/AIN1 or negative samples. The specificity for HSIL/AIN2–3 detection for a sensitivity established at 70% was 44.7% (95%confidence interval [CI] 36.5–53.2) for TOP2A and MKI67 and 54.5% (95%CI 46.0–62.8%) for CDKN2A/p16. mRNA detection of cell biomarkers in anal liquid-based cytology is feasible. Further studies are warranted to confirm if strategies based on mRNA detection have any role in anal cancer screening.

## 1. Introduction

Although anal cancer is infrequent in the general population, over the past decade its incidence and mortality have increased by about 2.2% and 2.9% per year, respectively [1]. The incidence of anal cancer is mainly concentrated in some specific, well-defined groups of patients, including men who have sex with men, particularly those living with human immunodeficiency virus (HIV), other non-HIV-related immunosuppressed populations, and women with human papillomavirus (HPV)-associated disease [2,3,4,5]. Despite these groups have different anal cancer risk, overall, the incidence of this neoplasia exceeds that of cervical cancer, the overall most frequent HPV-associated cancer [6]. Thus, there is a growing interest in developing screening and surveillance strategies for anal cancer, targeting these high-risk populations.

Many analogies have been recognized between anal and cervical cancer. Indeed, the common etiology of these two cancers (HPV) has prompted the use of the screening strategies performed for cervical cancer to prevent anal cancer. However, there are important differences between these two diseases that need to be addressed in the populations at risk [2,3,4]: the diverse evolution of the two diseases in the absence of treatment (natural history) [7,8] and the results in terms of disease control of the clinical management of the two premalignant lesions (clinical outcomes) [1]. 

Although scant data is available, the screening strategies currently available for the prevention of anal cancer and its precursor, high-grade squamous intraepithelial lesion/anal intraepithelial neoplasia grade 2–3 (HSIL/AIN2–3) are far from ideal. As a result, the proposed screening recommendations are mainly based on expert opinions. Anal cytology has shown a low sensitivity and specificity for HSIL/AIN2–3 detection [1], whereas HPV testing provides an excellent sensitivity but has an extremely poor specificity in high-risk populations [9,10,11]. 

The detection of host-cell mRNA biomarkers involved in DNA replication and cell cycle progression, such as TOP2A, and other HPV-induced cell molecules involved in cell cycle control, such as MKi67 or p16 (CDKN2A), has proved to be feasible in cervical samples preserved in liquid-based cytology media [12]. Moreover, previous studies have shown that these biomarkers are potentially useful in the detection of intraepithelial lesions of the uterine cervix. Preliminary results suggest that this strategy might be useful for the detection of cervical HSIL/CIN2–3 [12]. However, there are no data on the use of mRNA-based techniques to detect these host-cell biomarkers in anal liquid-based cytology samples. The large size of the area to be sampled due to the anatomical complexity of the anal canal, the keratinizing nature of the epithelium which frequently causes the exfoliation of a limited number of viable cells, together with the presence of contaminating fecal material [13,14,15] result in samples that are technically challenging to analyze. Consequently, few studies have been performed using molecular methods. [16,17]

In the present study we aimed to determine the feasibility of the detection of the mRNA of three HPV-associated biomarkers (TOP2A, MKi67 and CDKN2A) in anal samples preserved in liquid-based cytology and to evaluate whether this approach provides valuable information for the detection of HSIL/AIN2–3. 

## 2. Materials and Methods

### 2.1. Study Design and Case Selection

For this cross-sectional study, anal cytology samples, collected in the Anal Cancer Prevention Unit from January to December 2016 from men who have sex with men living with HIV, were retrieved from the Department of Pathology of the Hospital Clinic. All patients had been referred to the Anal Cancer Prevention Unit from the HIV Unit of our center. All were under highly active antiretroviral therapy (HAART) and had an undetectable viral load and CD4 counts > 400 cells/mL. Patients with anal sample fulfilling the following inclusion criteria were considered eligible for study: (1) anal liquid-based-cytology sample with adequate material for conventional cytology and HPV testing; (2) in cases with abnormal cytology and/or positive HPV test, a high-resolution anoscopy with at least one directed biopsy [18].

For this validation study we considered the following three diagnostic categories as concordant results: (a) negative (control group): normal anal cytology result and negative HPV test; (b) LSIL: anal cytology result of LSIL and biopsy confirming LSIL/AIN1; and (c) HSIL/AIN: anal cytology result of HSIL and biopsy confirming HSIL/AIN2–3 [12,19]. The following were considered as exclusion criteria: (1) previous history or histological diagnosis of anal cancer in the simultaneous biopsy; (2) previous treatment for HSIL/AIN2–3 performed within the previous 3 years; and (3) previous HPV vaccination.

A total of 215 patients who fulfilled the inclusion criteria were included in the study: 70 with a negative result (control group), 70 with LSIL/AIN1 (LSIL group) and 75 with HSIL/AIN2–3 (HSIL group). The study was approved by the institutional ethical review board of Hospital Clínic of Barcelona (2.26/01/2018, 16 February 2018) and all patients provided signed informed consent.

### 2.2. Cytology Sampling, High-Resulution Anoscopy and Biopsy

The anal sample was collected using a moistened Dacron swab, which was transferred to PreservCyt solution (Hologic, Marlborough, MA, USA). In all patients with abnormal cytology a high-resolution anoscopy was performed. Prior to the anoscopy, a digital anorectal examination was carried out to rule out abnormalities suggestive of anal cancer. The high-resolution anoscopy was performed using an Olympus Evis Exera II CV-180 anoscope (Olympus, Barcelona, Spain). Anal evaluation was carefully performed with 5% acetic acid, and iodine-based solution was used to stain the anal canal and to highlight areas of abnormal epithelium. On the detection of suspicious lesions, a high-resolution anoscopy directed biopsy was obtained.

### 2.3. Liquid-Based Cytology, Histological Diagnosis and HPV Testing

The first part of the anal sample was used for cytological analysis. The residual material was used first for HPV testing and thereafter for RNA isolation. 

Thin-layer cytology slides were prepared using the Thinprep T2000 slide processor (Hologic) and stained using the Papanicolaou method. Cytology slides were evaluated by a cytotechnologist (L.M.) and reviewed by a citopathologist (L.R.) using the revised Bethesda criteria [20]. Subsequently, liquid-based cytology samples were used for mRNA evaluation.

All histological samples were fixed in 10% formalin-fixed and embedded in paraffin following routine procedures. Four-µm sections were routinely stained with hematoxylin and eosin (HandE). All the histological samples were carefully reviewed by a pathologist (J.O.) with experience in HPV-related lesions to confirm the presence or absence of SIL/AIN and its grade. The histological diagnoses were established using pure morphologic criteria based on the HandE-stained sections, with no knowledge of HPV status or the cytology result [21]. Biopsy specimens were classified as normal, LSIL/AIN1, and HSIL/AIN2–3 according to the lower anogenital squamous terminology (LAST) criteria [22]. p16 immunohistochemical staining was performed in all the cervical samples obtained. A positive block staining for p16 in the dysplastic area was required for the diagnosis of HSIL/AIN2–3 [23].

HPV detection was performed using the Xpert HPV Assay (Xpert; Cepheid, Sunnyvale, CA) in the material collected in liquid-based media (PreservCyt) [24]. The Xpert HPV Assay (Cepheid, Inc, Sunnyvale, CA) is a quantitative polymerase chain reaction (qPCR) assay that uses disposable cartridges able to detect 14 types of high-risk HPV DNA (types 16, 18, 31, 33, 35, 39, 45, 51, 52, 56, 58, 59, 66, 68). Xpert HPV assay results offer partial genotyping including HPV 16, HPV 18 and/or 45, and high-risk HPV other than 16, 18 or 45.

### 2.4. RNA Isolation, Reverse Transcriptase PCR (RT-PCR) and qPCR

Once the cytological exam and the HPV testing had been performed, 5mL of the cell suspension was centrifuged at 3000G for 5 min, and the cell pellet was mixed with 700 μL QIAzol lysis reagent (Qiagen, Hilden, Germany). Total RNA was extracted using the RNeasy RNA extraction kit (Qiagen), according to the manufacturer’s protocol. Sample preparation was carried out with the highest measures of quality control to avoid contamination and cross-contamination. RNA concentrations were measured with a Nano-Drop instrument (Thermo Scientific, Wilmington, DE, USA). The total RNA yield varied between 0.5 and 64.3 mg, reflecting the variability of the cell numbers in the specimens.

Reverse transcription (RT) was performed starting with 10 μL of total RNA in a 20 μL reaction volume using random hexamers and the high capacity cDNA RT-kit from Applied Biosystems (Foster City, CA, USA) according to the manufacturer’s protocol. To exclude DNA contamination a reaction without RT was run in parallel with each specimen, as described previously [12].

All PCRs were performed in triplicate at a reaction volume of 25 μL containing 5 μL of cDNA diluted at 1:5 and mixed with Taqman Universal PCR MasterMix (Applied Biosystems). The following protocol was used for all assays: denaturation (10 min at 95 °C) and amplification (15 sec at 95 °C, 1 min at 60 °C) repeated for 40 cycles. The expression of CDKN2A/p16, MKI67 and TOP2A, was analyzed. Beta glucuronidase (GUSB) and cGMP-dependent protein kinase 1 (PKG1) were selected as reference genes for quality control of the RNA. This combination of reference genes showed a high stability in expression between groups of normal samples versus HSIL samples. The SiHa cell line was selected as a reference for RNA normalization.

The number of cycles required for the signal to cross the threshold (cycle threshold [Ct] value) for target genes was set at 35 cycles [12]. For the reference gene PRKG1, a Ct value above 35 cycles indicates poor RNA quality. Samples above these Ct values were therefore considered invalid and excluded from the analysis. The primer and probe sequences used in the qPCR are shown in Table 1 [12]. All probes were FAM-labeled, and all reactions were run in singleplex.

### 2.5. Data Analysis

The data were analyzed with the SPSS program (Version 24.0). The analysis of variance test was used to compare quantitative variables between the different categories. The χ2 test was used for comparisons between categorical variables. A p ≤ 0.05 was considered statistically significant. 

All data related to expression are presented as ΔΔCt, which was calculated as (the Ct value of the target gene − the Ct value of the reference gene) − (the Ct value of the target gene for SiHa − the Ct value of the reference gene for SiHa) [25,26]. SiHa cells were used as reference, as these are HPV16-immortalized squamous cells, considered as a model for HPV-induced alterations associated with cell immortalization. [27,28] High ΔΔCt values correspond to low expression of a marker, whereas a low ΔΔCt corresponds to a high expression of that marker. The level of significance between target gene expression in normal samples and HSIL/AIN2–3 samples was analyzed using ROC analysis. Using ROC curves, we obtained the specificity and ΔΔCt cut off value of the different biomarkers for a sensitivity of 70%, 80% and 90%.

## 3. Results

One-hundred and ninety-nine out of 215 (92.6%) samples were considered adequate for analysis, and only 16 could not be evaluated due to low expression of the reference genes and/or low RNA yield. Of the 199 samples finally included in the study 66 (33.2%) were normal, 66 (33.2%) LSIL, and 67 (33.7%) HSIL. No differences in terms of adequacy for analysis were observed between the three groups of samples (66/70 [94.3%] normal, 66/70 [94.3%] LSIL, and 67/75 [89.3%] HSIL, *p* = 0.419). 

Table 2 and Figure 1 show the mRNA levels of TOP2A, MKI67 and CDKN2A/p16 expressed as ΔΔCt [(Ct target-Ct reference sample) − (Ct SiHa-Ct reference SiHa)] in each diagnostic group. The mRNA expression was significantly higher in HSIL compared with LSIL and negative samples for the three biomarkers: TOP2A (*p* = 0.043 and *p* = 0.030, respectively), MKI67 (*p* = 0.011 and *p* < 0.001, respectively) and CDKN2A/p16 (*p* = 0.011 and *p* < 0.001, respectively). CDKN2A/p16 expression in LSIL specimens was also significantly higher compared with negative samples (*p* = 0.030), but no differences were observed between LSIL and negative samples for the other two biomarkers.

The ROC curves and the area under curve for the identification of patients with HSIL/AIN2–3 for the different biomarkers are shown in Figure 2 (A–C). The specificity and the cutoff values for the identification of HSIL/AIN2–3 for sensitivity established at 70%, 80% and 90% is shown in Table 3. The combination of the different biomarkers did not significantly improve the sensitivity and/or specificity of the tests for the identification of patients with HSIL/AIN2–3.

Figure 3 shows the HPV genotyping results in each diagnostic group. HPV16 was significantly more frequent in the HSIL group compared with the LSIL group (*p* = 0.003). Multiple HPV genotypes were found in 61.7% (63/199) of the samples studied. The prevalence of multiple infection was more frequent in the HSIL than in the LSIL group (56.7 [38/67] vs. 37.9%; [25/66]; *p* = 0.037). 

Table 4 and Figure 4 show the ΔΔCt for the different biomarkers according to the HPV genotype. The expression of the three biomarkers (TOP2A, MKI67 and CDKN2A/p16) was significantly higher in samples which were positive for the HPV16 genotype compared with HPV18 and/or 45 or high-risk HPV other than 16, 18 or 45. Patients with multiple infections showed a significantly higher expression of the three biomarkers (TOP2A, MKI67 and CDKN2A/p16) compared with patients with a single infection. ΔΔCt values in multiple and single infections were respectively 3.26 (95% confidence interval 2.60–3.92) and 3.98 (95% confidence interval 3.61–4.35) for TOP2A (*p* = 0.044); 3.22 (95% confidence interval 2.66–3.79) and 4.12 (95% confidence interval 3.73–4.50) for MKI67 (*p* = 0.010) and 0.95 (95% confidence interval 0.44–1.46) vs. 1.84 (95% confidence interval CI 1.52–2.16) for CDKN2A/p16 (*p* = 0.003). Considering only multiple infections, although the expression of TOP2A, MKI67 and CDKN2A/p16 was higher in patients with multiple infections harboring HPV16 than in those not harboring this genotype, the differences were only statistically significant for CDKN2A/p16.

## 4. Discussion

This is the first study evaluating the mRNA expression of host-cell genes involved in anal HPV-associated carcinogenesis in men who have sex with men living with HIV. Our results show that this evaluation may be feasible in anal liquid-based cytology samples in a clinical setting. Thus, molecular assays designed to detect the mRNA of host-cell biomarkers in specimens preserved in liquid-based cytology media may be a promising approach for HSIL/AIN2–3 detection. 

The current screening options for anal lesions in men who have sex with men living with HIV have shown not only poor reproducibility, but also limited sensitivity and specificity to detect anal HSIL/AIN2–3 [1,29]. On the basis of World Health Organization recommendations, several European guidelines have endorsed screening based on HPV testing rather than cytology for cervical cancer, the most common HPV-associated neoplasia, which has many analogies with anal cancer. [30] However, the utility of anal cancer screening based on HPV testing in populations at risk is limited due to its poor specificity, since anal HPV infection is detected in over 80% of these individuals [7] [31,32]. 

In the present study, we evaluated the feasibility of mRNA detection and its possible usefulness for HSIL/AIN2–3 detection. Due to the technical difficulties of the anal cytology samples, we initially focused on the technical issues. Extraction of high-quality RNA is a prerequisite for the application of RNA transcripts as biomarkers for clinical purposes [12,33]. Few reports have analyzed the value of biomarkers from anal scrapes, which have been considered as being particularly challenging because of the low quality (attributed to a low cell count and fecal contamination) of some samples [29]. Remarkably, in our study, almost 93% of the specimens had RNA of adequate quality, which could be successfully analyzed. These reliable results might be related to the high accuracy of the storage and sample processing, which are critical steps to ensure high-quality RNA and show that, when performed and processed appropriately, anal cytology samples preserved in liquid-based cytology fixatives are adequate for RNA evaluation. The satisfactory results obtained in the feasibility evaluation are in keeping with previous studies performed in cervical cancer [12,34] and indicate that evaluate this approach for anal cancer screening may be clinically interesting. 

The biomarkers analyzed in this study have shown good reproducibility, sensitivity, and specificity in other HPV-associated malignancies such as cervical cancer. A previous study performed by our group suggested that the analysis of mRNA expression of the same host-cell genes analyzed in this study (TOP2A, MKI67 and CDKN2A/p16) is not only feasible but also useful for the detection of cervical HSIL/CIN2–3 [12]. In our previous study, TOP2A, MKI67 and CDKN2A/p16 showed sensitivities of 97%, 93% and 75%, respectively for HSIL/CIN2–3 with a specificity of 54%, 56% and 78%, respectively. In contrast with the promising results observed in the cervix, these biomarkers have shown suboptimal sensitivity and specificity for detecting HSIL/AIN2–3 in anal samples [31]. Indeed, the specificity of the different biomarkers for a sensitivity of 70%, which could be considered as the minimum value acceptable for screening purposes, was 44.7% for TOP2A and MKI67 and 54.5% for CDKN2A/p16. Moreover, an increased sensitivity implied an important decrease in the specificity, suggesting these biomarkers are not adequate for screening purposes or that, if they were used for screening of HSIL/AIN2–3, additional tests would be required to improve sensitivity and specificity. Unfortunately, the selection criteria of the present study, designed to evaluate the feasibility of mRNA expression in anal samples, do not allow proper comparison of the accuracy to detect HSIL/CIN2–3 between different tests such as cytology or HPV testing or the combinations of tests

In the present study HPV16 was the most frequent genotype identified in anal precursors, and its prevalence was significantly higher in HSIL/AIN2–3 compared with LSIL/AIN1. These results are in keeping with previous reports showing that HPV16 is the most common genotype as an etiologic factor in HSIL/AIN2–3 and anal cancer [1,11,35]. As in previous studies, in the present series HPV16 was positive in about 60% of the HSIL/AIN2–3 lesions and HPV18 was found in 7.5% of the HSIL/AIN2–3 samples. [36,37]. It has been shown that the percentage of HPV16 and 18 genotypes increases with the severity of the diseases, and it is particularly high in invasive cancers [38]. Thus, HPV16 and/or 18 genotyping alone or in combination with additional tests might also be useful within an anal cancer screening strategy [29,39,40]. Multiple HPV infections are also commonly reported in the anal region. Indeed, the number of HPV types has been strongly associated with increasing levels of histological abnormality [41,42] which is consistent with the results of this study. 

In the present study, we observed that the mRNA expression of host-cell genes was particularly high in HPV16-positive samples. No previous studies have evaluated the relationship between the expression of host-cell genes and different HPV genotypes in cytological anal samples. Thus, this is the first study to evaluate the relation between HPV genotypes and mRNA expression of host-cell genes in anal cytological samples from men having sex with men with HIV and SIL/AIN. Our results suggest that this combination might be an interesting approach in risk-based management anal cancer screening, simulating the approach that has recently been suggested for cervical screening [1]. Interestingly, the mRNA expression of host-cell genes was higher in patients with multiple infections compared with those with a single infection. In other HPV-related premalignant lesions, such as in the uterine cervix, previous studies have shown the expression of the viral oncogene E6/E7 is higher in multiple HPV infections compared with single infections [43]. However, there are no previous data about the effect of multiple infections on the expression of host-cell genes in the anal canal in men having sex with men living with HIV. Interestingly, in the subset of patients with multiple infections, those involving HPV16 showed higher mRNA expression of the host-cell genes than those without HPV16. Although the differences were only significant for CDKN2A/p16, the trend was consistent for all the genes evaluated, reinforcing the importance of HPV genotyping in anal cancer screening. 

This study has some limitations. The possible changes in the levels of expression of the mRNA of the biomarkers in patients with HPV infection not associated with intraepithelial lesions were not evaluated. In our clinical setting, following the recommendations for anal cancer screening, this particular high-risk population of men who have sex with men living with HIV with a negative cytology does not undergo further examination with high-resolution anoscopy, and consequently, none of these patients had histological confirmation of the negative result. Thus, the control group had to be recruited based only on the cytology result, and we included the additional criteria of a negative HPV test to ensure that the patients were truly negative and not false negative results of the cytology. Nevertheless, the main aim of our study was to determine the feasibility of performing mRNA analysis of host-cells in a standard clinical setting, which was independent of the patients selected. Another limitation is that the presence and effect of low-risk HPV types could not be evaluated with the genotyping test used in our study. However, although it has been suggested that low-risk HPV types may have a role in the appearance of anal lesions, the risk of low-risk HPV infection in the development and progression of anal premalignancies is clearly lower than the risk due to high-risk HPV infection [44,45].

In conclusion, we have shown that the determination of mRNA expression in anal liquid-based cytology specimens is feasible. Further studies including a larger number of genes and patients representing the complete picture found in an actual screening program are warranted to confirm that strategies based on mRNA detection of cell biomarkers might have a role in the secondary prevention of anal cancer.

## Figures and Tables

**Figure 1 diagnostics-09-00173-f001:**
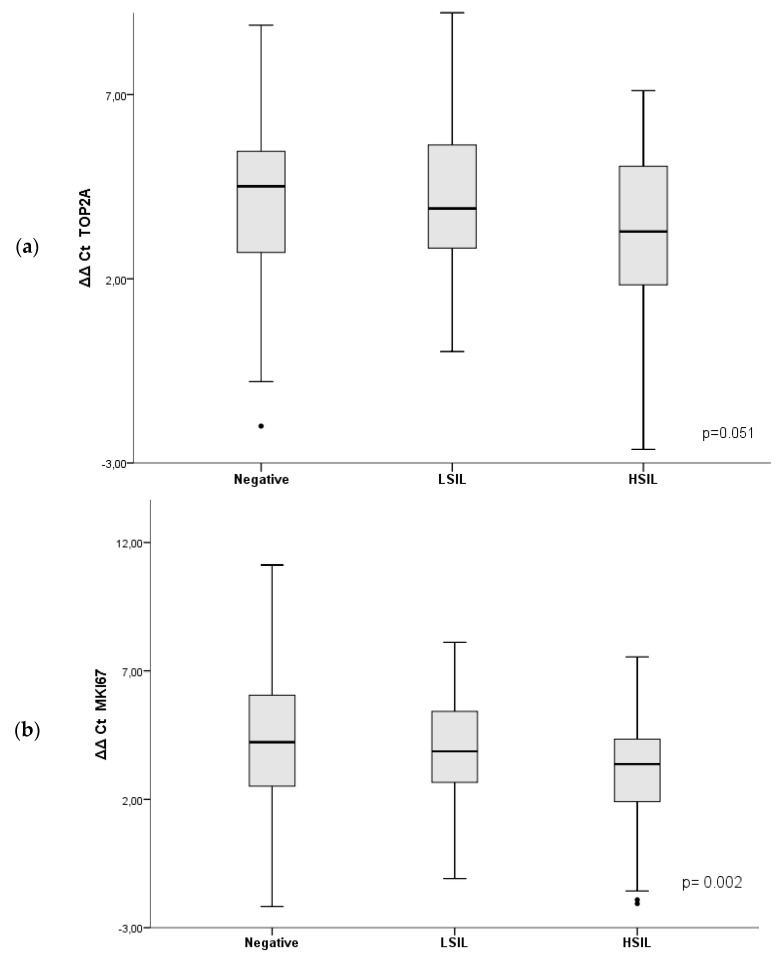
Boxplots showing the mRNA levels of (**a**) TOP2A; (**b**) MKI67; (**c**) CDKN2A/p16 in the different diagnostic groups. Patients were grouped into three categories according to the cytological, histological, and virological results. These categories included: (a) Negative (negative Pap test result and negative high-risk human papillomavirus [HPV] test); (b) low-grade squamous intraepithelial lesion (LSIL) (Pap test result of LSIL and biopsy showing anal intraepithelial neoplasia grade 1 [LSIL/AIN1]); and (c) high-grade squamous intraepithelial lesion (HSIL) (patients with Pap test result of HSIL and biopsy confirming HSIL/AIN2–3). The values shown are expressed as ΔΔCt [(cycle threshold target-cycle threshold reference sample) − (cycle threshold SiHa-cycle threshold reference SiHa)]. The black line within the box represents the median; the whiskers represent the minimum and maximum values that lie within 1.5 interquartile ranges from the end of the box. Values outside this range are represented by black dots.

**Figure 2 diagnostics-09-00173-f002:**
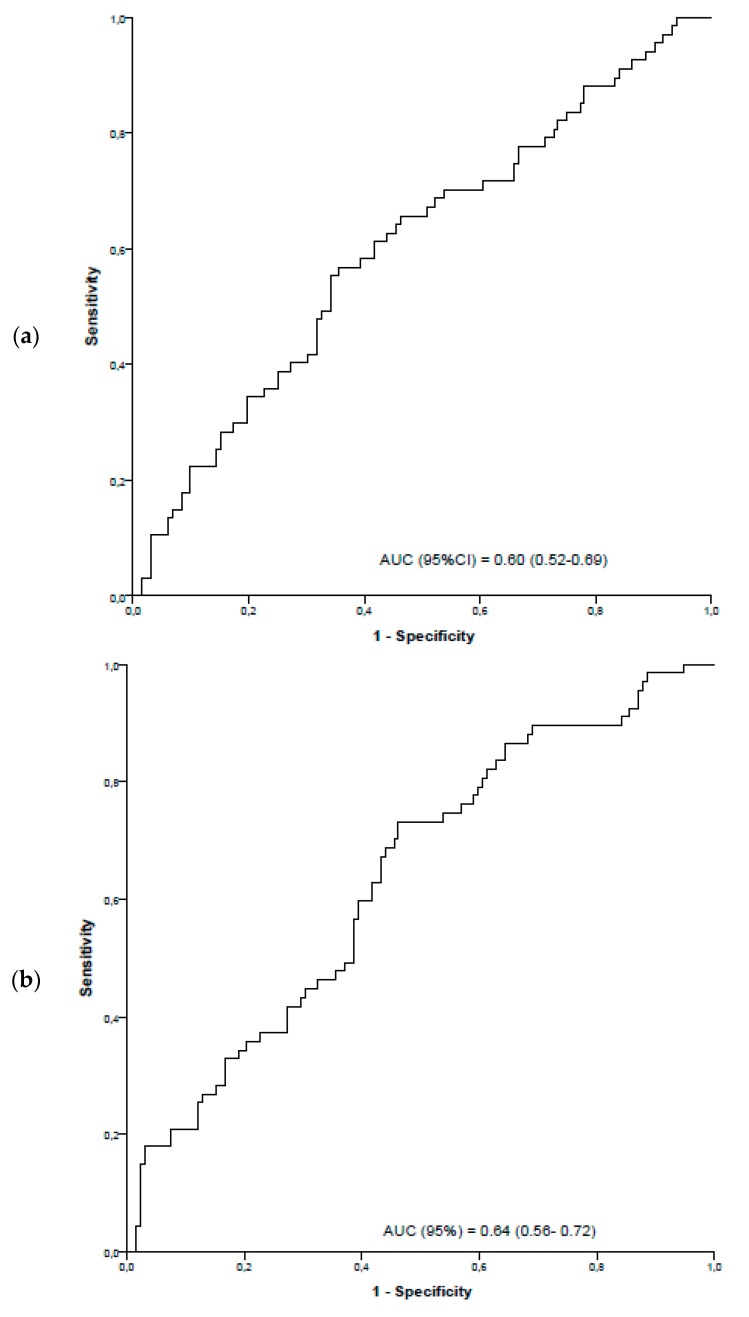
ROC curves and the area under curve for the identification of patients with HSIL/AIN2–3. mRNA levels are presented as ΔΔCt [(cycle threshold target-cycle threshold reference sample) − (cycle threshold SiHa-cycle threshold reference SiHa)]. The values of the areas under the ROC curve (AUC) and the 95% confidence intervals (95% CI) are presented. (**a**) TOP2A; (**b**) MKI67; (**c**) CDKN2A/p16.

**Figure 3 diagnostics-09-00173-f003:**
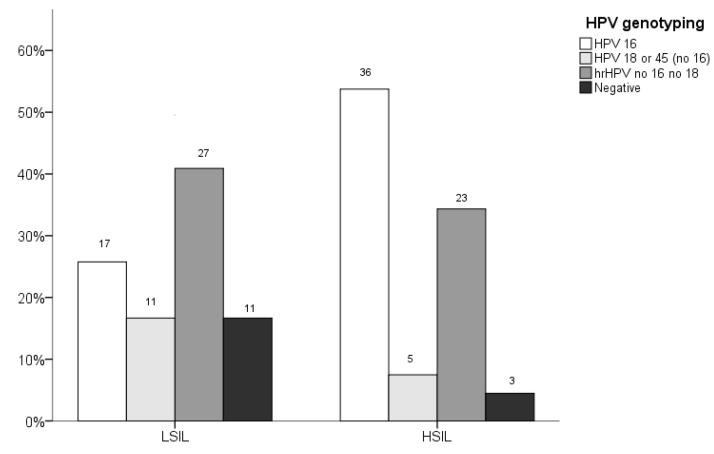
Human papillomavirus (HPV) genotyping in the patients with anal lesions. The criteria for the group of low-grade squamous intraepithelial lesion (LSIL) were: Pap test result of LSIL and biopsy showing anal intraepithelial neoplasia grade 1 [LSIL/AIN1]); the criteria for high-grade squamous intraepithelial lesion (HSIL) were: Pap test result of HSIL and biopsy confirming HSIL/AIN2–3). The results are presented in absolute numbers. Bars represent the percentage of HPV genotypes within the diagnostic group.

**Figure 4 diagnostics-09-00173-f004:**
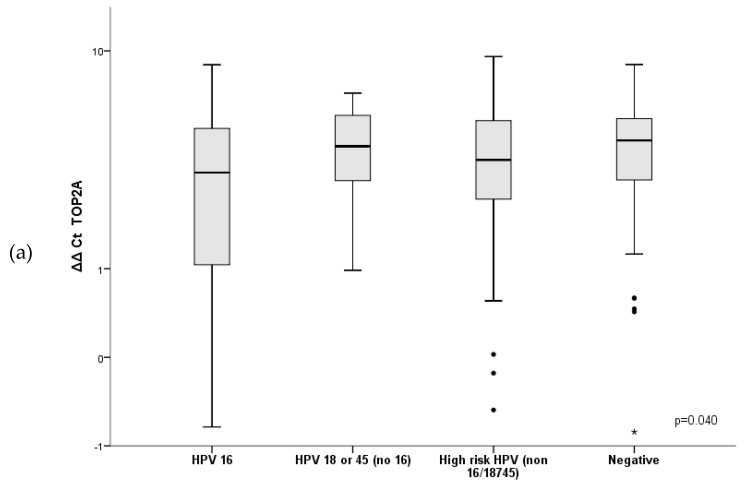
Boxplots of the levels of (**a**) TOP2A; (**b**) MKI67; (**c**) CDKN2A/p16 according to the HPV genotype. The values shown are ΔΔCt [(cycle threshold target-cycle threshold reference sample) − (cycle threshold SiHa-cycle threshold reference SiHa)]. The black line within the box represents the median; the whiskers represent the minimum and maximum values that lie within 1.5 interquartile ranges from the end of the box. Values outside this range are represented by black dots.

**Table 1 diagnostics-09-00173-t001:** Primers and probes used to detect the mRNA expression of the biomarkers analyzed in the study.

Target Gene		Source
*CDKN2A*	F: 5′-CATAGATGCCGCGGAAGGT-3′	Life Technologies
	R: 5′-CCCGAGGTTTCTCAGAGCCT-3′	
	P: FAM-CCTCAGACATCCCCGATTGAAAGAACC-TAMRA	
*MKI67*	MKI67 (Hs010332443_m1	Life Technologies
*TOP2A*	TOP2A (Hs03063307_m1)	Life Technologies
*GUSB*	GUSB (Hs99999908_m1)	Life Technologies
*PGK1*	PGK1 (Hs99999906_m1)	Life Technologies

**Table 2 diagnostics-09-00173-t002:** mRNA levels of TOP2A, MKI67 and CDKN2A/p16 according to the diagnostic group.

	Diagnosis						
	Negative		LSIL		HSIL		*p*
TOP2A	4.07	(3.50–4.65)	4.00	(3.44–4.57)	3.19	(2.62–3.75)	0.051
MKI67	4.44	(3.81–5.07)	3.99	(3.52–4.46)	3.08	(2.54–3.61)	0.002
CDKN2A/p16	2.32	(1.85–2.80)	1.59	(1.13–2.06)	0.77	(0.33–1.21)	<0.001

The values shown are expressed as ΔΔCt [(cycle threshold target-cycle threshold reference sample) − (cycle threshold SiHa-cycle threshold reference SiHa)] and 95% confidence interval. Patients were grouped into three categories according to the cytological, histological, and virological results. These categories included: (a) negative (negative Pap test result and negative high-risk human papillomavirus [HPV] test); (b) low-grade squamous intraepithelial lesion (LSIL) (Pap test result of LSIL and biopsy showing anal intraepithelial neoplasia grade 1 [LSIL/AIN1]); and (c) high-grade squamous intraepithelial lesion (HSIL) (patients with Pap test result of HSIL and biopsy confirming HSIL/AIN2–3).

**Table 3 diagnostics-09-00173-t003:** Cut-off values and specificity of TOP2A, MKI67 and CDKN2A/p16, for the diagnosis of histologically confirmed high-grade squamous intraepithelial lesion (HSIL/AIN2–3) for sensitivity established at 70%, 80% and 90%.

	Sensitivity (%)	Cut-Off	Specificity (%)	(95% CI)
**TOP2A**	70	4.50	44.7	(36.5–53.2)
	80	6.01	20.7	(14.8–28.2)
	90	7.11	11.4	(7.2–17.8)
**MKI67**	70	4.01	44.7	(36.5–53.2)
	80	6.40	20.7	(14.8–28.2)
	90	7.60	10.7	(6.6–16.9)
**CDKN2A/p16**	70	1.77	54.5	(36.5–53.2)
	80	2.98	30.7	(23.7–38.8)
	90	5.24	10.0	(6.0–16.1)

**Table 4 diagnostics-09-00173-t004:** mRNA levels of TOP2A, MKI67 and CDKN2A/p16 according to the HPV genotype. The values shown are ΔΔCt [(cycle threshold target-cycle threshold reference sample) − (cycle threshold SiHa-cycle threshold reference SiHa)] and 95% confidence interval.

	HPV16		HPV18 or 45		High-Risk HPV (non 16/18/45)		Negative		*p*
TOP2A	2.99	(2.23–3.75)	4.20	(3.24–5.15)	3.82	(3.24–4.40)	4.12	(3.63–4.62)	0.040
MKI67	3.17	(2.54–3.80)	3.61	(2.43–4.78)	3.70	(3.13–4.26)	4.40	(3.87–4.94)	0.020
CDKN2A/p16	0.74	(0.21–1.26)	2.05	(0.95–3.16)	1.26	(0.76–1.76)	2.20	(1.77–2.62)	<0.001

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
