# Peer review of "mRNA Detection in Anal Cytology: A Feasible Approach for Anal Cancer Screening in Men Who Have Sex with Men Living With HIV"

_diagnostics, 2019, doi:10.3390/diagnostics9040173_

Round 1
Reviewer 1 Report
Authors have presented a manuscript entitled ´mRNA Biomarker in Anal Cytology: A Feasible Approach for Anal Cancer Screening in Men Who Have Sex with Men Living With HIV´ to be considered for publication in the journal Diagnostics.
This manuscript aims to emphasize the feasibility of mRNA biomarkers from anal liquid cytology samples for anal cancer screening. The topic is timely and the synthesis of the manuscript is good, in general. The work builds on previously published data and discusses the known challenges in HPV-related anal cancer diagnosis. Presented results show that the used mRNA biomarkers, except TOP2A, have a moderate resolution to detect anal HSILs from LSILs (MKi67 AUC 0.64, CDKN2A 0.68). Moreover, the authors show significant differences in mRNA biomarker expression levels between HPV16, HPV18/HPV45 or high-risk HPV infected subjects. However, regarding multiple infected cases, which is high also in their dataset, the authors should more clearly stratify and reanalyze their data. Indeed, a detailed dissemination of possible expressions level difference could be analyzed between single and multiple oncogenic HPVs infected cases, particularly between HPV16 single and HPV16 multiple infected cases. This because, multiple HR-HPVs have shown more oncogenic E6/E7 HPV mRNA than single infections in previous studies. Hence, the effect of multiple HPVs infections in anal cancer etiology, especially among HIV-positives, should be discussed in the manuscript. And low-risk HPVs occasional role in HSILs also previously reported should be considered in the analysis and discussed. Finally the particularly challenging histological classification of HPV-related anal lesions should be well integrated and elaborated in the manuscript (See Clavero et al 2017 Papillomavirus Research).
Author Response
QUESTION 1:
“…regarding multiple infected cases, which is high also in their dataset, the authors should more clearly stratify and reanalyze their data. Indeed, a detailed dissemination of possible expressions level difference could be analyzed between single and multiple oncogenic HPVs infected cases, particularly between HPV16 single and HPV16 multiple infected cases. This because, multiple HR-HPVs have shown more oncogenic E6/E7 HPV mRNA than single infections in previous studies. Hence, the effect of multiple HPVs infections in anal cancer etiology, especially among HIV-positives, should be discussed in the manuscript. And low-risk HPVs occasional role in HSILs also previously reported should be considered in the analysis and discussed.”
RESPONSE:
We agree with the reviewer about the relevance of the controversial issue regarding the impact of multiple HPV infections in the development of HPV-related lesions. Thus, following his/her suggestion we have stratified and reanalyzed our data evaluating the impact of multiple HPV infection, particularly those containing the HPV 16 genotype.
As a result, a new paragraph has been added to the Results section in the revised manuscript. (page, 10 lines 272 to 281):
“Patients with multiple infections showed a significantly higher expression of the three biomarkers (TOP2A, MKI67 and CDKN2A/p16) compared with patients with a single infection. ΔΔ Ct values in multiple and single infections were respectively 3.26 (95% confidence interval 2.60-3.92) and 3.98 (95% confidence interval 3.61-4.35) for TOP2A (p=0.044); 3.22 (95% confidence interval 2.66-3.79) and 4.12 (95% confidence interval 3.73-4.50) for MKI67 (p=0.010) and 0.95 (95% confidence interval 0.44-1.46) vs. 1.84 (95% confidence interval CI 1.52-2.16) for CDKN2A/p16 (p=0.003). Considering only multiple infections, although the expression of TOP2A, MKI67 and CDKN2A/p16 was higher in patients with multiple infections harboring HPV16 than in those not harboring this genotype, the differences were only statistically significant for CDKN2A/p16 (data not shown).”
A new paragraph has also been added to the Discussion section of the revised manuscript. (page 13, lines 369 to 378):
“Interestingly, the mRNA expression of host-cell genes was higher in patients with multiple infections compared with those with a single infection. In other HPV-related premalignant lesions, such as in the uterine cervix, previous studies have shown the expression of the viral oncogene E6/E7 is higher in multiple HPV infections compared with single infections (Andersson et al., 2012). However, there are no previous data about the effect of multiple infections on the expression of host-cell genes in the anal canal in men having sex with men living with HIV. Interestingly, in the subset of patients with multiple infections, those involving HPV16 showed higher mRNA expression of the host-cell genes than those without HPV16. Although the differences were only significant for CDKN2A/p16, the trend was consistent for all the genes evaluated, reinforcing the importance of HPV genotyping in anal cancer screening.”
A new reference has also been added to the new version:
(Andersson et al., 2012): Andersson, E., Kärrberg, C., Rådberg, T., Blomqvist, L., Zetterqvist, B. M., Ryd, W., … Horal, P. (2012). Type-dependent E6/E7 mRNA expression of single and multiple high-risk human papillomavirus infections in cervical neoplasia. Journal of Clinical Virology, 54(1), 61–65. https://doi.org/10.1016/j.jcv.2012.01.012
Finally, the role of low-risk HPV types was not evaluated, since the genotyping test used for genotyping in our study does not provide information on low-risk HPV types. However, the objective of our study was to evaluate the feasibility of detecting mRNA in anal cytology. In addition, the high-risk HPV type is responsible for the large majority of sites harboring HPV-associated lesions. Thus, although we accept that this may be a limitation, we consider that it does not decrease the validity of the results presented in the present manuscript.
To stress this issue a new paragraph has been added to the Discussion section (page 13, lines 389 to 393):
“Another limitation is that the presence and effect of low-risk HPV types could not be evaluated with the genotyping test used in our study. However, although it has been suggested that low-risk HPV types may have a role in the appearance of anal lesions, the risk of low-risk HPV infection in the development and progression of anal premalignancies is clearly lower than the risk due to high-risk HPV infection (Donà et al., 2018; Leeman et al., 2019).”
Two new references have also been added to the new version:
(Donà et al., 2018): Donà, M. G., Benevolo, M., Latini, A., Rollo, F., Colafigli, M., Frasca, M., … Giuliani, M. (2018). Anal Cytological Lesions and HPV Infection in Individuals at Increased Risk for Anal Cancer. Cancer Cytopathology, 126(7), 461–470. https://doi.org/10.1002/cncy.22003
(Leeman et al., 2019): Leeman, A., Jenkins, D., Marra, E., van Zummeren, M., Pirog, E. C., van de Sandt, M. M., … Quint, W. G. V. (2019). Grading immunohistochemical markers p16 INK4a and HPV E4 identifies productive and transforming lesions caused by low‐ and high‐risk HPV within high‐grade anal squamous intraepithelial lesions. British Journal of Dermatology, bjd.18342. https://doi.org/10.1111/bjd.18342
QUESTION 2:
“…the particularly challenging histological classification of HPV-related anal lesions should be well integrated and elaborated in the manuscript (See Clavero et al 2017 Papillomavirus Research).”
RESPONSE:
We agree with the reviewer that the histological classification of HPV-associated anal lesions is particularly challenging. For that reason, a pathologist with lengthy experience in HPV-associated lesions carefully reviewed all the samples included in the study. Moreover, in order to minimize the misclassification of lesions, in addition to using the LAST classification, we included p16 immunostaining in all cases, and p16 positivity was required for the diagnosis of HSIL/AIN2-3 (see methods section, page 3, lines 130-131). These very strict criteria for the diagnosis of HSIL/AIN2-3 used in our study are in keeping with those recommended in the study mentioned by the reviewer.
Following the reviewer’s recommendation, we have modified a sentence in the Methods section (page 3, lines 125 to 127, underlined the newly added text):
“Four-µm sections were routinely stained with hematoxylin and eosin (H&E). All the histological samples were carefully reviewed by a senior pathologist (J.O.) with experience in HPV-related lesions to confirm the presence or absence of SIL/AIN and its grade.”
The reference suggested by the reviewer has been also added to the new version (page 3, line 133):
(Clavero et al., 2017): Clavero, O., McCloskey, J., Molina, V. M., Quirós, B., Bravo, I. G., de Sanjosé, S., … Pimenoff, V. N. (2017). Squamous intraepithelial lesions of the anal squamocolumnar junction: Histopathological classification and HPV genotyping. Papillomavirus Research. https://doi.org/10.1016/j.pvr.2016.12.001
Please see the attachment.

Reviewer 2 Report
In this article, the author has shown that the detection of mRNA expression in anal liquid-based specimens is feasible. The article has several limitations and needs to be revised across different sections of the manuscript.
The title of the manuscript could be changed to 'mRNA detection in anal cytology: A feasible approach....' as authors have not narrowed down to any gene/protein as a biomarker. Table 2, 4, and 5 should all be represented as graphs; this way, the data is more presentable and easy to understand. At the bottom of each ROC curve, the author mention DDCt and gene name can author explain the reason to have such a label. Are all the enrolled patients HIV positive as there is no exclusion or inclusion criteria based on HIV status, viral load or CD44 count, with or without treatment for same.
Author Response
QUESTION 1: The title of the manuscript could be changed to 'mRNA detection in anal cytology: A feasible approach....' as authors have not narrowed down to any gene/protein as a biomarker.
RESPONSE:
Following the reviewer’s suggestion, we have modified the title of the manuscript, which in the revised version reads as follows (underlined the newly added word and crossed out the deleted word):
“mRNA Biomarkers detection in Anal Cytology: A Feasible Approach for Anal Cancer Screening in Men Who Have Sex with Men Living With HIV”
QUESTION 2: Table 2, 4, and 5 should all be represented as graphs; this way, the data is more presentable and easy to understand.
RESPONSE:
We agree with the reviewer that data are easier to understand when represented as graphs and thank him/her for this suggestion. Three new figures have been added to the revised version of the manuscript showing data of Tables 2, 4 and 5 (see the revised version of the manuscript):
“Figure 1. Boxplots showing the mRNA levels of (a) TOP2A; (b) MKI67; (c) CDKN2A/p16 in the different diagnostic groups. Patients were grouped into three categories according to the cytological, histological, and virological results. These categories included: a) negative (negative Pap test result and negative high-risk human papillomavirus [HPV] test; b) low-grade squamous intraepithelial lesion (LSIL) (Pap test result of LSIL and biopsy showing anal intraepithelial neoplasia grade 1 [LSIL/AIN1]); and c) high-grade squamous intraepithelial lesion (HSIL) (patients with Pap test result of HSIL and biopsy confirming HSIL/AIN2-3). The values shown are expressed as ΔΔ Ct [(cycle threshold target-cycle threshold reference sample) - (cycle threshold SiHa-cycle threshold reference SiHa)]. The black line within the box represents the median; the whiskers represent the minimum and maximum values that lie within 1.5 interquartile ranges from the end of the box. Values outside this range are represented by black dots.”
“Figure 3. Human papillomavirus (HPV) genotyping in the patients with anal lesions. The criteria for the group of low-grade squamous intraepithelial lesion (LSIL) were: Pap test result of LSIL and biopsy showing anal intraepithelial neoplasia grade 1 [LSIL/AIN1]); the criteria for high-grade squamous intraepithelial lesion (HSIL) were: Pap test result of HSIL and biopsy confirming HSIL/AIN2-3). *The negative group has been excluded because according to the inclusion criteria all patients included in this group were HPV-negative. The results are presented in absolute numbers. Bars represent the percentage of HPV genotypes within the diagnostic group.”
“Figure 4. Boxplots of the levels of (a) TOP2A; (b) MKI67; (c) CDKN2A/p16 according to the HPV genotype. The values shown are ΔΔCT [(cycle threshold target-cycle threshold reference sample) - (cycle threshold SiHa-cycle threshold reference SiHa)]. The black line within the box represents the median; the whiskers represent the minimum and maximum values that lie within 1.5 interquartile ranges from the end of the box. Values outside this range are represented by black dots.”
However, because the tables provide more detailed information about the data of mRNA expression, we have maintained Tables 2 and 5 showing the precise ΔΔ Ct values and 95% confidence intervals of TOP2A, MKI67 and CDKN2A/p16 according to the diagnostic group. Table 4 has been deleted and substituted by Figure 3.
To adjust the manuscript to the Figures added, a sentence has been modified in the Results section (page 5, line 191, underlined the newly added text):
“Table 2 and Figure 1 shows the mRNA levels of TOP2A, MKI67 and CDKN2A/p16 expressed as ΔΔCt [(Ct target-Ct reference sample) - (Ct SiHa-Ct reference SiHa)] in each diagnostic group.”
The former figure 1 is now figure 2 in the revised manuscript.
A sentence has been modified in the Results section (page 9, line 248, underlined the newly added text and crossed out the deleted word):
“Table 4 Figure 3 shows the HPV genotyping results in each diagnostic group.”
A sentence has been modified in the Results section (page 10, line 269, underlined the newly added text and crossed out the deleted word):
“Table 5 4 and Figure 4 show the ΔΔ Ct for the different biomarkers according to the HPV genotype.”
QUESTION 3: At the bottom of each ROC curve, the author mentions DDCt and gene name can author explain the reason to have such a label.
RESPONSE:
This was a mistake in the Figure. DDCt meant ΔΔ Ct (Ct value of the target gene - Ct value of the reference gene) - (Ct value of the target gene for SiHa - Ct value of the reference gene for SiHa), as suggested by previous studies (Rao, Huang, Zhou, & Lin, 2013; Schmittgen et al., 2000). This has been corrected in the revised version.
In any case, in order to clarify how the data were obtained in the ROC curves, we have added, , a new sentence to the legend of Figure 2 in the revised version (underlined the newly added text)
“Figure 2. ROC curves and the area under curve for the identification of patients with HSIL/AIN2-3. mRNA levels are presented as ΔΔ Ct [(cycle threshold target-cycle threshold reference sample) - (cycle threshold SiHa-cycle threshold reference SiHa)]. The values of the areas under the ROC curve (AUC) and the 95% confidence intervals (95% CI) are presented. (a) TOP2A; (b) MKI67; (c) CDKN2A/p16.”
QUESTION 4: Are all the enrolled patients HIV positive as there is no exclusion or inclusion criteria based on HIV status, viral load or CD44 count, with or without treatment for same.
RESPONSE:
As stated in the Methods section (Study Design and Case Selection), the anal cytology samples included in the studies were collected in the Anal Cancer Prevention Unit from men who have sex with men living with HIV. All patients were being followed in the HIV Unit of our center. All were under HAART treatment and had an undetectable viral load and normal CD4 count.
We have added all this information to the revised manuscript (page 3, lines 90 to 92, underlined the newly added text):
“For this cross-sectional study, anal cytology samples collected in the Anal Cancer Prevention Unit from January to December 2016 from men who have sex with men living with HIV were retrieved from the Department of Pathology of the Hospital Clinic. All patients had been referred to the Anal Cancer Prevention Unit from the HIV Unit of our center. All were under highly active antiretroviral therapy (HAART) and had an undetectable viral load and CD4 counts > 400 cells/mL.”
Please see the attachment

Reviewer 3 Report
The study by Marta del Pino et al demonstrated the utility and feasibility of mRNA (TOP2A, MKI67 and CDKN2A/p16) biomarkers to distinguish HSIL/AIN2-3. The results look interesting and promising. I’d like to have the authors address the following points for a further consideration.
Major points:
Line 85-90: Are you referring to any publicly available guidelines for classification? If yes, please specify and cite; if no, please put down in details how you define each.
Line 91-94: Have you obtained written informed consents from each of the participants? If yes, please write it down in this section. Ethical concern is an important issue in clinical study.
Line 144: Please provide justification on why you chose SiHa cell line as reference.
Line 248-261: How would you compare the sensitivity and specificity of your mRNA-based method with the existing method in detecting HSIL/CIN2-3 in anal samples? Also, as you have mentioned ‘additional tests would be required to improve sensitivity and specificity’, have you tried combining your mRNA-based method with other med testing results to see if you can achieve a better sensitivity and specificity?
Line 248-261: There are literature showing connection between HPV16 and your selected biomarkers (TOP2A, MKI67 and CDKN2A/p16). Please search, summarise and include them in this part since your HPV genotyping data demonstrate that these markers are associated with HPV16.
Minor points:
Line 46-47: The authors should specify what they mean by ‘in these population’ as they talked about a mixed population including both men and women in the previous text. I would not say it’s technically fair to compare to the incidence of cervical cancer in men. Please revise.
Line 52-53: What are the differences in natural history and clinical outcomes? Please include short texts on them in the paragraph. It is neither complete nor proper to open a topic without further introduction.
Line 108: Who is the cytotechnologist involved. Please mark down his/her initials.
Line 145-149: It’s good to see the quality control process of isolated RNA.
Figure 1: Please include ‘a’ ‘b’ ‘c’ in the figure panels. Also, please be consistent with gene names in the figure panels and legend.
Future direction: the authors can follow up these cases, especially among the HSIL/AIN2-3 group, to see if a higher expression of these markers would predict a higher rate of cancer events and related clinical outcomes.
Author Response
QUESTION 1: Line 85-90: Are you referring to any publicly available guidelines for classification? If yes, please specify and cite; if no, please put down in details how you define each.
RESPONSE:
Indeed, we are not referring to a classification but rather a way of selecting patients with unquestionable diagnosis that has previously been used in studies conducted by our group with similar objectives, performed in intraepithelial neoplasia of the uterine cervix, the standard model for HPV-related diseases. As stated in the paper, the main objective of the present work was to study the feasibility of mRNA detection and the conditions for sample collection, preservation and analysis required to obtain adequate results in anal samples, since there are no data on mRNA expression in this setting. Thus, in order to avoid the possible confounding effect caused by the inclusion of misclassified cases, we selected patients with a concordant cytological and histological diagnosis.
To clarify this issue, in the Methods section, we have added the references of other studies that have used these selection criteria for HPV-related lesions (page 3, line 100, underlined the newly added text):
“For this validation study we considered the following three diagnostic categories as concordant results: (a) negative (control group): normal anal cytology result and negative HPV test; (b) LSIL: anal cytology result of LSIL and biopsy confirming LSIL/AIN1; and (c) HSIL/AIN: anal cytology result of HSIL and biopsy confirming HSIL/AIN2-3 (del Pino et al., 2015; del Pino et al., 2019).”
A new reference has also been added to the new revised version of the manuscript:
“(del Pino et al., 2019): del Pino, M., Sierra, A., Marimon, L., Delgado, C. M., Rodriguez-Trujillo, A., Barnadas, E., … Ordi, J. (2019). CADM1, MAL, and mir124 promoter methylation as biomarkers of transforming cervical intrapithelial lesions. International Journal of Molecular Sciences. https://doi.org/10.3390/ijms20092262”
QUESTION 2: Line 91-94: Have you obtained written informed consents from each of the participants? If yes, please write it down in this section. Ethical concern is an important issue in clinical study.
RESPONSE:
We agree with the reviewer that ethical concern is an important issue in clinical studies. The present study was approved by the ethical review board of our institution, as stated in the Methods section, and all the patients had signed the informed consent.
Following the reviewer’s suggestion, this information has been added to the revised version of the manuscript (page 3, lines 106 to 107, underlined the newly added text):
“The study was approved by the institutional ethical review board of our institution (2.26/01/2018, 16 February 2018) and all patients provided signed informed consent.”
QUESTION 3: Please provide justification on why you chose SiHa cell line as reference.
RESPONSE:
SiHa has been used as a reference sample for ΔΔCt calculation (difference in ΔCt ((Ct target gene) - Ct (reference gene)) between the target and reference sample) as SiHa is an HPV16-immortalized squamous cell that has been largely considered as the reference cell for the studies of HPV-induced alterations associated with cell immortalization. To clarify this issue a new sentence has been added to the Methods section (page 4 and 5, lines 177 to 179, underlined the newly added text):
All data related to expression are presented as ΔΔ Ct, which was calculated as (the Ct value of the target gene - the Ct value of the reference gene) - (the Ct value of the target gene for SiHa - the Ct value of the reference gene for SiHa) (Rao, Huang, Zhou, & Lin, 2013; Schmittgen et al., 2000). SiHa cells were used as reference, as these are HPV16-immortalized squamous cells, considered as a model for HPV-induced alterations associated with cell immortalization. (de Campos et al., 2018; Yue et al., 2004)
In addition, two new references have been added to the new revised version of the manuscript:
(de Campos et al., 2018): de Campos, R. P., Schultz, I. C., de Andrade Mello, P., Davies, S., Gasparin, M. S., Bertoni, A. P. S., … Wink, M. R. (2018). Cervical cancer stem-like cells: systematic review and identification of reference genes for gene expression. Cell Biology International. https://doi.org/10.1002/cbin.10878
(Fu et al., 2018): Fu, J., Cheng, J., Liu, X., Li, J., Wei, C., Zheng, X., … Fu, J. (2018). Evaluation genotypes of cancer cell lines HCC1954 and SiHa by short tandem repeat (STR) analysis and DNA sequencing. Molecular Biology Reports. https://doi.org/10.1007/s11033-018-4438-7
QUESTION 4: Line 248-261: How would you compare the sensitivity and specificity of your mRNA-based method with the existing method in detecting HSIL/CIN2-3 in anal samples? Also, as you have mentioned ‘additional tests would be required to improve sensitivity and specificity’, have you tried combining your mRNA-based method with other med testing results to see if you can achieve a better sensitivity and specificity? .
RESPONSE:
As stated in the manuscript, the data currently available on the impact of screening strategies in anal cancer are very scarce. The common etiology of anal and cervical cancers has prompted the implementation of the screening tools used for cervical cancer (i.e. cytology and HPV testing), which have been used for many years to prevent anal cancer. However, these proposed screening recommendations are mainly based on expert opinions and not on the results of clinical trials or epidemiological studies. Different studies have shown that anal cytology has a low sensitivity and specificity for the detection of anal HSIL/AIN2-3 (Clarke & Wentzensen, 2018), whereas while HPV testing provides an excellent sensitivity, it has an extremely poor specificity in high-risk populations (Drabeni, Clemente, Moise, Bon, & Fontana, 2015; Lin, Franceschi, & Clifford, 2018; Nowak et al., 2016).
We agree with the reviewer that comparison of the sensitivity and specificity of mRNA detection with combined approaches including other screening tests such as anal cytology and/or HPV testing) would be of interest. However, before these comparisons are conducted, it should first be proven that these new tools, which could improve the results of the existing tests (in our study, the detection of the mRNA of three HPV-associated biomarkers, TOP2A, MKi67 and CDKN2A/p16), are feasible in anal samples and that evaluating these biomarkers provides valuable information for the detection of HSIL/AIN2-3. Thus, in order to minimize the risk of evaluating the mRNA status of anal samples that had been over or underdiagnosed, we included only samples with concordant cytological-histological results. These strict inclusion criteria as well as the study design hinder the possibility of evaluating the sensitivity of the anal cytology or HPV testing, the tests mainly proposed for anal cancer screening.
To clarify this issue, we have added a new sentence in the Discussion section (page 12, lines 344 to 347, underlined the newly added text):
” Moreover, an increased sensitivity implied an important decrease in the specificity, suggesting these biomarkers are not adequate for screening purposes or that, if they were used for screening of HSIL/AIN2-3, additional tests would be required to improve sensitivity and specificity. Unfortunately, the selection criteria of the present study, designed to evaluate the feasibility of mRNA expression in anal samples, do not allow proper comparison of the accuracy to detect HSIL/CIN2-3 between different tests such as cytology or HPV testing or the combinations of tests.”
QUESTION 5: Line 248-261: There are literature showing connection between HPV16 and your selected biomarkers (TOP2A, MKI67 and CDKN2A/p16). Please search, summarise and include them in this part since your HPV genotyping data demonstrate that these markers are associated with HPV16.
RESPONSE:
A number of studies have addressed the analysis of mRNA expression of the host-cell genes involved in carcinogenesis in the uterine cervix. These studies mostly analyzed the relationship between the expression of the biomarkers TOP2A, MKi67 and CDKN2A/p16 and the grade of the HPV-related lesion. However, as stated in the Discussion section, as far as we know, “no previous studies have evaluated the relationship between the expression of host-cell genes and different HPV genotypes in cytological anal samples “. To stress the novelty of the results presented in the present manuscript, a new sentence has been added to the Discussion section (page 13, lines 364 to 366, underlined the newly added text):
“In the present study, we observed that the mRNA expression of host-cell genes was particularly high in HPV16-positive samples. No previous studies have evaluated the relationship between the expression of host-cell genes and different HPV genotypes in cytological anal samples. Thus, this is the first study to evaluate the relation between HPV genotypes and mRNA expression of host-cell genes in anal cytological samples from men having sex with men with HIV and SIL/AIN.”
QUESTION 6: Line 46-47: The authors should specify what they mean by ‘in these population’ as they talked about a mixed population including both men and women in the previous text. I would not say it’s technically fair to compare to the incidence of cervical cancer in men. Please revise.
RESPONSE:
As stated in the text, men who have sex with men, particularly those living with human immunodeficiency virus (HIV), other non-HIV-related immunosuppressed populations, and women with human papillomavirus (HPV)-associated disease are considered high-risk groups for anal cancer. Despite being heterogeneous groups and the fact that the prevalence of anal cancer differs slightly in the different groups, all have a significantly higher risk of anal cancer compared with the general population. Following the reviewer’s suggestion, to clarify this issue, we have added a new sentence to the Introduction section (page 2, lines 47 to 48, underlined the newly added text):
“The incidence of anal cancer is mainly concentrated in some specific, well-defined groups of patients. The highest frequency in decreasing order is observed in men who have sex with men, particularly those living with human immunodeficiency virus (HIV), other non- HIV-related immunosuppressed populations, and women with human papillomavirus (HPV)- associated disease (Bregar et al., 2018; D’Souza et al., 2008; Robison et al., 2015; Silverberg et al., 2012). In these populations, Despite these groups have different anal cancer risk, overall, have an increased the risk of this neoplasia anal cancer exceeds that of cervical cancer, the overall most frequent HPV-associated cancer6.”
QUESTION 7: Line 52-53: What are the differences in natural history and clinical outcomes? Please include short texts on them in the paragraph. It is neither complete nor proper to open a topic without further introduction.
RESPONSE:
Following the recommendation of the reviewer, a short text has been included in the Introduction section to clarify all these concepts (page 2, lines 56 to 59, underlined the newly added text and crossed out the deleted word):
However, there are important differences between these two diseases that need to be addressed in the populations at risk (Bregar et al., 2018; D’Souza et al., 2008; Silverberg et al., 2012): the diverse evolution of the two diseases in the absence of treatment (natural history) of the two diseases (Machalek, Poynten, & Jin, 2012; Nicolas Wentzensen & Clarke, 2017) and the results in terms of disease control of the clinical management of the two premalignant lesions (clinical outcomes) (Clarke & Wentzensen, 2018).
QUESTION 8: Line 108: Who is the cytotechnologist involved. Please mark down his/her initials.
RESPONSE:
Following the reviewer’s recommendation, the initials of the cytotechnologists involved in the study have been indicated (page 3, line 122, underlined the newly added text):
Cytology slides were evaluated by a cytotechnologist (L.M.) and reviewed by a cytopathologist (L.R.) using the revised Bethesda criteria (Solomon et al., 2002). Subsequently, liquid-based cytology samples were used for mRNA evaluation.
QUESTION 9: Line 145-149: It’s good to see the quality control process of isolated RNA.
RESPONSE:
The sample preparation was carried out with the highest measures of quality control to avoid contamination and cross-contamination. In addition, as already stated in the subsection “RNA isolation, reverse transcriptase PCR (RT-PCR) and qPCR” of the Methods section (page 4, lines 152 to 154): “To exclude DNA contamination a reaction without RT was run in parallel with each specimen, as described previously (del Pino et al., 2015).”
A new sentence has been added to the Methods section to emphasize this issue (page 4, lines 145 to 146, underlined the newly added text):
“Total RNA was extracted using the RNeasy RNA extraction kit (Qiagen), according to the manufacturer’s protocol. Sample preparation was carried out with the highest measures of quality control to avoid contamination and cross-contamination. RNA concentrations were measured with a Nano-Drop instrument (Thermo Scientific, Wilmington, DE, USA). The total RNA yield varied between 0.5 and 64.3 mg, reflecting the variability of the cell numbers in the specimens.”
QUESTION 10: Figure 1: Please include ‘a’ ‘b’ ‘c’ in the figure panels. Also, please be consistent with gene names in the figure panels and legend.
RESPONSE:
Following the reviewer’s suggestion, we had included “a”, “b”, and “c” in Figure 2 of the revised version of the manuscript panel. We have also changed the gene name CDKN2A to CDKN2A/p16 to be consistent with the rest of the manuscript (underlined the newly added text):
“Figure 2. ROC curves and the area under curve for the identification of patients with HSIL/AIN2-3. mRNA levels are presented as ΔΔ Ct [(cycle threshold target-cycle threshold reference sample) - (cycle threshold SiHa-cycle threshold reference SiHa)] The values of the areas under the ROC curve (AUC) and the 95% confidence intervals (95% CI) are presented. (a) TOP2A; (b) MKI67; (c) CDKN2A/p16.”
Please see the attachment

Round 2
Reviewer 2 Report
Authors have addressed all concerns of the reviewer carefully. The reviewer has no further comments, and the manuscript could be accepted in present form.
Author Response
The reviewer says: "The reviewer has no further comments, and the manuscript could be accepted in present form."
Reviewer 3 Report
Thank you for the revision. I'm happy to see the improvements in the manuscript. Further to Question One, I assume it'll be helpful for the readers to know what your Negative, LSIL/AIN1, and HSIL/AIN2-3 groups are like by adding in-house representative cytological and histological images.
Author Response
QUESTION 1: Further to Question One, I assume it'll be helpful for the readers to know what your Negative, LSIL/AIN1, and HSIL/AIN2-3 groups are like by adding in-house representative cytological and histological images.
RESPONSE:
Following the reviewer suggestion a new Figure has been added shouwing a representative cytological and histological images of each diagnostic group
Figure 1. Images of the cytological and histological samples for each diagnostic group. (A and A’) Control group: normal anal cytology; (B and B’) LSIL group: anal cytology (B) and biopsy (B’); (C and C’) HSIL group: anal cytology (C) and biopsy (C’)
(page 3, line 100 and 101, underlined the newly added text):
“For this validation study we considered the following three diagnostic categories as concordant results: (a) negative (control group): normal anal cytology result and negative HPV test; (b) LSIL: anal cytology result of LSIL and biopsy confirming LSIL/AIN1; and (c) HSIL/AIN: anal cytology result of HSIL and biopsy confirming HSIL/AIN2-3 (del Pino et al., 2015; del Pino et al., 2019). Figure 1 shows an example of the cytological and histological samples for each diagnostic category.”
The former figure 1, 2 , 3 and 4 are now figure 2, 3, 4 and 5 in the revised manuscript
